# Current State of Molecular Cytology in Thyroid Nodules: Platforms and Their Diagnostic and Theranostic Utility

**DOI:** 10.3390/jcm13061759

**Published:** 2024-03-19

**Authors:** Zeina C. Hannoush, Roberto Ruiz-Cordero, Mark Jara, Atil Y. Kargi

**Affiliations:** 1Division of Endocrinology, Diabetes and Metabolism, Department of Internal Medicine, Miller School of Medicine, University of Miami, Miami, FL 33136, USA; maj158@med.miami.edu; 2Department of Pathology and Laboratory Medicine, Miller School of Medicine, University of Miami, Miami, FL 33136, USA; rxr1314@med.miami.edu; 3Department of Neurosurgery, University of North Carolina at Chapel Hill, Chapel Hill, NC 27599, USA; akargi@email.unc.edu

**Keywords:** thyroid cancer, fine needle aspiration, thyroid nodule, molecular cytology, genetic analysis, mutation

## Abstract

The high prevalence of thyroid nodules and increased availability of neck ultrasound have led to an increased incidence of diagnostic thyroid fine needle aspirations, with approximately 20% yielding indeterminate results. The recent availability of molecular tests has helped guide the clinical management of these cases. This paper aims to review and compare three main commercially available molecular cytology platforms in the U.S.—Afirma GSC, Thyroseq GC, and ThyGeNEXT + ThyraMIR. Sequential improvements of the Afirma GSC and Thyroseq GC tests have increased positive and negative predictive values, sensitivity, and specificity. Comparative studies revealed similar diagnostic performance between these tests, with considerations for factors such as cost and processing time. Thyroseq GC provides detailed genomic information and specific management recommendations. ThyGeNEXT + ThyraMIR, though less studied, presents promising results, particularly in miRNA analysis for weak driver mutations. Challenges in interpreting results include variations in reporting and the evolving nature of testing platforms. Questions persist regarding cost-effectiveness and the utility of ultrasound characteristics in selecting candidates for molecular testing. While molecular testing has primarily served diagnostic purposes, advancements in understanding genetic alterations now offer therapeutic implications. FDA-approved options target specific genetic alterations, signaling a promising future for tailored treatments.

## 1. Introduction 

Thyroid nodules are highly prevalent and detectable in up to 25% of the general population; they are easily detectable by neck ultrasound, which is a noninvasive diagnostic imaging modality that has become more readily accessible in the last 20 years. The rise in the detection of thyroid nodules has led to an increase in diagnostic thyroid fine needle aspirations (FNA). This widespread occurrence has been paralleled by a relatively stable death rate for thyroid cancer since 2009. For the most common and well-differentiated types of thyroid malignancy, the survival rate can be remarkably high, reaching 97% [1]. 

About 20% of the cytology results from thyroid FNA come back as an indeterminate result, which includes atypia or follicular lesion of indeterminate significance (Bethesda III) (AUS/FLUS) and follicular neoplasm/suspicious for follicular or oncocytic (formerly Hurthle) neoplasm (Bethesda IV) [2]. Approximately 10 to 30% of these nodules will ultimately yield a malignant pathology [2]. This situation places significant financial strain on healthcare systems, payers, and patients alike [3]. For many years, there has been a significant need for accurate tools that can help identify which of these nodules carry a malignant pathology from those with indolent behavior without the need for invasive lobectomy or thyroidectomy procedures [4]. In recent years, different molecular cytology platforms have become available to help guide clinical management in these patients. In this paper, we aim to review and compare the performance of the three main platforms currently commercially available in the U.S. 

## 2. Background

Before the recent emergence of molecular cytology applied to thyroid FNA, former American Thyroid Association (ATA) guidelines indicated that when a thyroid FNA resulted in an indeterminate diagnosis, the clinician could either repeat the FNA, indicate a thyroid lobectomy or continue surveillance with a follow-up ultrasound [5]. In approximately 60–70% of such instances, repeating the FNA will reclassify the nodule to a different Bethesda category, and in approximately 50%, it will result in a benign diagnosis, which has a greater NPV than molecular tests [6]. Further strategies available to help guide management in such indeterminate cases included seeking a second cytopathology opinion and correlating the cytology findings with the sonographic characteristics of the nodule [7]. 

With the advent of various molecular cytology platforms becoming more readily available, the 2015 ATA management guidelines indicate that “for nodules with AUS/FLUS cytology, after consideration of worrisome clinical and sonographic features, investigations such as repeat FNA or molecular testing may be used to supplement malignancy risk assessment in lieu of proceeding directly with a strategy of either surveillance or diagnostic surgery. Informed patient preference and feasibility should be considered in clinical decision-making (weak recommendation, moderate-quality evidence). If repeat FNA cytology and/or molecular testing are not performed or are inconclusive, either surveillance or diagnostic surgical excision may be performed for an AUS/FLUS thyroid nodule, depending on clinical risk factors, sonographic pattern, and patient preference (strong recommendation, low-quality evidence)” [5].

In these guidelines, clinical judgment plays a significant role, and healthcare professionals need to consider various fac-tors before making decisions. Some information we know regarding ultrasound or molecular profile for all thyroid nodules might not necessarily apply to the subgroup of nodules with indeterminate cytology. There are certain nuances that apply specifically to this group that need careful consideration [8]. For instance, more than 70% of ATA high suspicion nodules on sonography end up being malignant, with most of those yielding a Bethesda category VI result on FNA; however, when we look at the nodules that have yielded a Bethesda category III result or indeterminate cytology, the predictive value of ultrasound decreases significantly. Similarly, we know that the *BRAF* V600E mutation is highly prevalent in papillary thyroid cancers, with its reported frequency varying from 30 up to 85% among studies [9]; however, when looking at the cancers with Bethesda category III and IV cytology, we find a lower prevalence of BRAF mutations and a higher presence of other mutations such as RAS mutations [10]. 

When considering the use of molecular testing to risk stratify a thyroid nodule with indeterminate cytology, we should aim for a test with a high negative predictive value (NPV) and high positive predictive value (PPV) to help better triage patients for thyroid surgery. Ideally, performance would be similar to a Bethesda II (benign) and Bethesda VI (malignant) cytology, where there are only 3% false positive and false negative rates [11]. 

When evaluating the currently available molecular testing options, several important factors should be considered. These include the number of studies conducted to evaluate the performance of each test, the duration of follow-up of patients, the percent of patients that underwent the gold standard diagnostic test, which is histopathological surgical confirmation, whether they included Bethesda category III, IV, and/or V as indeterminate cytology, as well as the type of study: prospective vs. retrospective, single vs. multicenter, and blinded vs. unblinded. A notion of the prevalence of malignancy in the tested population vs. the rate of malignancy in an institution is also very important, as this can affect the negative and positive predictive value of the test. In contrast, sensitivity and specificity are inherent and unchangeable characteristics that changes in the prevalence of the disease would not modify.

Of note is that there has been a significant evolution in the information provided by the different molecular tests since they first became commercially available. Initial reports were binary, either positive or negative, with a clear negative and positive predictive value. With a better understanding of the different mutations follicular thyroid cells can acquire and the different risks for malignancy each of these can convey, current reports usually provide either a benign or negative result vs. a positive prognostic value with different types of positive results depending on the specific mutation identified. This level of granularity in the report makes it more challenging to calculate a general positive predictive value for each test. An additional point of consideration in the evaluation of these tests is the variations present in the classification and reporting as a positive or negative molecular test result of premalignant neoplasms such as noninvasive follicular thyroid neoplasm with papillary-like nuclear features (NIFTP) and the performance of the tests in oncocytic neoplasms. 

## 3. Currently Commercially Available Molecular Tests for Indeterminate Thyroid Nodule Cytology

In the U.S., currently, there are three commercially available molecular tests in use for diagnosis of indeterminate thyroid nodule cytology: Afirma GSC, which is an RNA-based genomic sequencing classifier (GSC) using gene expression to infer the malignant potential of a nodule; ThyGeNext + ThyraMIR, which is a multiplatform test that involves an assessment of genetic alterations of DNA, mRNA, and gene expression regulators known as micro RNA; and Thyroseq GC, a multigene genomic classifier that started as a purely DNA test but in recent years has also added mRNA testing for a gene expression classifier. 

### 3.1. Afirma GSC

When a sample is sent for Afirma testing, it is first run through initial classifiers to identify if the sample contains parathyroid origin cells or rare neoplasms and lesions with >95% risk of malignancy (strong driver mutations), such as RET mutations causing medullary thyroid cancer, BRAF, and RET:PTC fusion. Then, a follicular content index is run, which identifies samples with sufficient molecular content. The follicular content index measures the relative abundance of follicular cells in the biopsy sample, which can affect the accuracy and interpretation of the GSC result. The specific details of how the FCI is calculated may be proprietary to the Afirma GSC test and may not be publicly disclosed by Veracyte. In general, the FCI is determined using a combination of gene expression patterns associated with follicular cells and other cell types present in the thyroid nodule. The expression levels are then normalized by the expression levels of other reference genes that are not affected by follicular content. The normalized expression levels are then summed to obtain the follicular content index. The higher the follicular content index, the more follicular cells are present in the sample. If there is not enough material, the results will be reported as insufficient. Following this, the ensemble classifier leverages multiple machine learning algorithms to derive a final benign vs. suspicious result, and lastly, a Hürthle Classifier analysis has been added to better classify Hürthle/oncocytic cell lesions as benign or malignant [12]. 

The *Afirma GSC* classifier identifies about one-third of indeterminate cytology samples as suspicious, a result that carries an overall 50% risk of malignancy (ROM). Of those, 44% have an identifiable alteration (variant/fusion) that can be detected with the most recently developed Afirma Expression Atlas (XA), comprised of 593 genes, 905 variants, and 235 fusion pairs, significantly increasing their ROM and providing more granular data that can guide clinical management. 

Reviewing the literature available to assess the performance of the Afirma test, one can conclude that about two-thirds of the Afirma *GSC* tests yield a negative result. The sensitivity and NPV vary between 90 and 100%, while some studies have suggested a lower NPV [13]. The PPV is around 60%, which is significantly improved compared to the 40% seen in the older version of the Afirma GEC test [14]. When looking at the clinical utility of the test, results show that 33 to 40% of tested patients underwent surgery, and about 60% or more of the tested patients will be able to avoid unnecessary surgery by using this test [15,16,17,18,19,20,21,22,23,24,25,26].

It is important to highlight that when discussing nodules with premalignant histopathology that are classified as benign, such as follicular adenomas and NIFTP, these benign lesions and their malignant counterparts, such as follicular carcinoma and follicular variants of PTC, often have the same molecular test result profiles. It is likely that in many cases, it might just be a matter of time before a “benign” premalignant nodule acquires additional genomic alterations that lead to invasion and thus fulfill criteria for malignancy. It is worth mentioning that Afirma suspicious nodules that are benign on final histopathology are more often clonal neoplasms such as NIFTP or follicular adenoma (FA); in contrast, Afirma benign nodules are more often hyperplastic nodules without premalignant potential. If we consider that neoplasms such as FAs could be considered premalignant and surgical removal is the appropriate treatment, the “false positive” rate of the Afirma test would be much lower [27].

### 3.2. Thyroseq 

Molecular testing of DNA from indeterminate cytology thyroid nodules for cancer mutations has had multiple modifications through the years, with progressive improvement in its sensitivity. Usually, with improvements in sensitivity, there tends to be a loss of specificity, though fortunately, this has not been the case with this test [28]. The procedure of Thyroseq testing in its most recent version involves several steps. First, there is an assessment of DNA and RNA adequacy for testing; this is followed by a cellular composition determination, for example, medullary thyroid cancer or parathyroid tissue. Then, next-generation sequencing (NGS) analysis is conducted for four classes of genetic alterations in 112 genes: (i) mutations (>12,000 variants), (ii) gene fusions (>150 types), (iii) copy number alterations, and (iv) gene expression alterations. The results are processed by a proprietary genomic classifier, and finally, the test result interpretation is based on a knowledge database of >3000 cases with known surgical outcomes, allowing for an assessment of cancer probability and risk of cancer recurrence [29].

Ultimately, reports of the Thyroseq test provide a negative or positive result, and the positive results provide a probability of malignancy or NIFTP (if applicable) as well as very detailed genomic information and clinical management guidance recommendations. A negative result carries a ROM of 3 to 4%, and simple observation is recommended. A positive result is subclassified into different categories: Cases with mutations in genes that carry a ROM of <10% are classified as currently negative. This group is most commonly comprised of NIFTP or neoplastic hyperfunctioning nodules with indolent behavior for which active surveillance is recommended. Nodules with RAS-like mutations or gene expression alterations (GEA) that carry a ROM of 30 to 80% are classified as positive RAS-like and recommended to undergo thyroid lobectomy. Nodules with oncocytic morphology that harbor copy number alterations (CNA) with a ROM of 40 to 80% are classified as positive Hürthle cell type with an intermediate ROM and are recommended to undergo either lobectomy or total thyroidectomy. For nodules classified as positive intermediate-risk, based on the presence of BRAF-like mutations or GEA, total thyroidectomy or lobectomy is recommended based on a ROM ranging from 95 to 100%. Lastly, for nodules classified as positive high-risk, based on mutations associated with a 98 to 100% ROM, total thyroidectomy with or without cervical lymph node dissection is recommended. 

The *Thyroseq* test performance has been validated in many studies. In one large prospective, double-blind study, the sensitivity and specificity were relatively high, with a negative predictive value of 97% and a positive predictive value of 66% [29]. Some other studies have shown a lower positive predictive value depending on how they group the mutations. In terms of clinical utility, the use of this test seems to help avoid about 61% of unnecessary surgeries [21,30,31].

A study looking at the Thyroseq test’s performance by specific histopathology type showed it was able to accurately predict 11/11 NIFTP, 24/27 papillary thyroid cancer (PTC), 21/22 follicular variant (FV), 10/10 Hürthle cell carcinomas, 3/4 follicular carcinomas, 1/1 medullary thyroid cancer, and 1/1 metastatic carcinoma [29].

### 3.3. ThyGeNEXT/ThyraMIR

ThyGeNext/ThyraMIR is a two-step test, and the first step is the ThyGeNEXT panel that tests for 10 gene mutations and 38 gene fusions, responsible for the most commonly occurring malignancies. Step 2 is the ThyraMIR, which tests the expression of 10 miRNA genes. The miRNA analysis may be valuable in predicting the behavior of nodules with weak driver mutations such as RAS, but further studies should be conducted to prove this concept [32]. A retrospective study evaluating the impact of pairwise miRNA expression analysis on risk stratification of 178 indeterminate thyroid nodules found significant improvement in the diagnostic accuracy of the test [33].

ThyGeNext/ThyraMIR has a three-tier reporting system that is either negative, moderate, or positive. A study analyzing this test’s performance showed that in the negative result group, only 4 out of 81 samples turned out to be malignant vs. 35 out of 47 in the positive result group. The moderate group has a risk of malignancy that is similar to the pretest risk of malignancy, and in the validation cohort, a moderate result occurred in 28% of patients. The test had a benign call rate of 46%, and positive results were found in 26%. Since this is a nonbinary test, the interpretation of the moderate risk category in calculating PPV and NPV has a major effect on these predictive parameters. The test can be sent on one dedicated pass, slides, or cell blocks and does not require refrigeration [32].

## 4. Comparison of Molecular Tests and Cost-Effectiveness Considerations 

Most of the available studies conducted to compare the performance of different molecular tests provide data comparing the performance of Afirma vs. Thyroseq tests with limited comparison data for the ThyGeNEXT/ThyraMIR test against the others. 

An independent, head-to-head test comparison study conducted at UCLA performed a prospective parallel randomized trial of 372 Bethesda III-IV nodules that were randomized to Afirma GSC (n = 201) or ThyroSeq v3 (n = 171) test. The results showed that the diagnostic performance of both tests was high with no statistical difference; there was a lower percentage of inadequate samples for Thyroseq 4% vs. 9% with Afirma and a higher benign call rate for Thyroseq 60% vs. 53% with Afrima [21]. A meta-analysis that included 12 validation and real-world experience studies that reported the performance of Thyroseq (530 nodules) and Afirma (471 nodules) also showed comparable results [28]. Other studies comparing both tests have also found very similar performances between the two tests in terms of NPV, PPV, benign call rate, and clinical utility [12,29,34,35].

Of note, the detailed report that Thyroseq provides, describing the specific molecular alterations found, including the variant allele frequency for mutations, can add valuable information to help clinical management decisions. Ultimately, as highlighted in one of the aforementioned studies’ conclusions, the choice of molecular test to be used may hinge on factors other than the diagnostic performance, such as cost, processing time, sample inadequacy rate, and information regarding specific mutations that can guide future treatment [21]. For instance, the TERT promoter mutation is a well-known high-risk mutation that can be detected by the Thyroseq and the ThyGeNEXT/ThyraMIR tests but not by the *Afirma* test. 

ThyroSeq GC and Afirma GSC have similar processing times of 7 to 10 business days after the sample is received by the laboratory. The processing time for ThyGeNEXT/ThyraMIR may vary, but it usually takes around 2 to 3 weeks to receive results after the laboratory receives the FNA sample. ThyroSeq GC and Afirma GSC costs are comparable at several hundred to a few thousand dollars. The combined cost of ThyGeNEXT (genetic testing) and ThyraMIR (microRNA testing) is typically higher compared to the other tests. It can range from a few thousand to several thousand dollars. These tests are currently widely used in the United States and are almost always covered by commercial insurance. The cost of the tests will depend on the benefits available through insurance. Deductible, copay, or coinsurance may apply depending on the case. Medicare covers testing. Patients may be eligible for financial assistance programs and can apply through the manufacturer’s website. The tests have been studied for cost-effectiveness since avoiding unnecessary surgeries in up to 50% of indeterminate cytology nodules allows for far less cost to the healthcare system from using the tests, not to mention the potential costs of life-long levothyroxine replacement [36]. As highlighted by a comparative analysis published previously, considering the diagnostic accuracy, cost, and feasibility of available tests is crucial [37]. This is particularly relevant to low- and middle-income countries where these sophisticated tests might still not be readily available.

A summary of the key features of the three main commercially available molecular tests can be found in Table 1.

Many questions remain regarding the cost-effectiveness of these highly valuable tests and whether the ultrasound characteristics of thyroid nodules can be used to select which nodules with indeterminate cytology are better candidates for molecular testing. A study conducted at the University of Miami aimed to investigate this question and compared the performance of older versions of the Afirma and Thyroseq tests (Afirma GEC and Thyroseq v2). The results showed that the sonographic risk of thyroid nodules, classified by ATA-US and TIRADS guidelines, alone was not an adequate predictor of malignancy. There was a modest correlation of the sonographic risk category with molecular test results, and the NPV of both molecular tests was not altered by the sonographic risk category. While 75% of high sonographic risk nodules were Thyroseq v2 positive, the NPV remained high in this category. The PPV of Afirma GEC was higher in higher sonographic risk categories, while the PPV of Thyroseq was similar in nodules regardless of the sonographic risk category [13]. Another multicenter Thyroseq and ultrasound study found that neither the ATA nor TI-RADS scoring systems further informed the risk of cancer/NIFTP beyond that predicted by Thyroseq [38].

## 5. Therapeutic Implications for the Future

Molecular testing in thyroid nodule FNA has been primarily used for diagnostic purposes. Because of its high negative predictive value, many surgeries can be avoided in patients with indeterminate cytology and negative molecular testing. Recent advances in our understanding of the association between genetic alterations, cancer phenotype, and risk of aggressive behavior have allowed us to use the information provided by molecular tests to further guide management in different aspects:-Surgical Planning: For patients who undergo surgery, molecular testing can provide valuable information for surgical planning. It can help identify the extent of surgery needed, such as whether total thyroidectomy or lobectomy is appropriate, based on the risk of malignancy and the presence of specific genetic alterations.-Prognosis: Molecular testing can also provide prognostic information that helps predict the likelihood of recurrence or aggressive behavior of thyroid cancer. For instance, more aggressive tumors, such as differentiated high-grade thyroid cancer (DHGTC), a recently defined category from the 2022 WHO classification, are known to have exclusive characteristics, molecular patterns, and transcriptional profiles with higher BRAFV600E mutation and gene fusions rates [39]. This information is valuable for guiding postoperative management decisions, including the need for adjuvant therapy and the frequency of surveillance.-Monitoring Response to Therapy: Molecular testing can be used to monitor response to therapy in patients with advanced thyroid cancer. Changes in molecular profiles over time may indicate treatment response or the emergence of resistance mechanisms, guiding adjustments to treatment strategies.-Hereditary Risk Assessment: Molecular tests are able to identify germline mutations associated with hereditary thyroid cancer syndromes. Identification of these mutations allows for appropriate genetic counseling and screening of at-risk family members.-Treatment Selection: In cases of advanced or recurrent thyroid cancer, molecular testing may inform treatment decisions, such as the selection of targeted therapies or participation in clinical trials based on the presence of specific molecular alterations. Molecular testing can be used to assess the mRNA expression of sodium iodide symporter (NIS) in the sampled tissue and potentially predict the effectiveness of radioactive iodine (RAI) treatment [40]. Furthermore, molecular markers may assist in therapeutic decisions beyond RAI, although this practice has yet to be widely adopted [8]. The Food and Drug Administration has approved therapeutic options for the treatment of advanced BRAFV600E-mutated thyroid carcinomas, NTRK fusions, RET-mutated medullary thyroid carcinoma, and RET-fusion papillary thyroid cancer [29]. As advances in the field continue to evolve, it is predicted that further treatments targeting specific genetic alterations will continue to become available.

## 6. Conclusions

Molecular tests are valuable tools in the management of thyroid nodules with indeterminate cytology. Testing algorithms have evolved and improved rapidly in the last few years to improve specificity and PPV while maintaining high sensitivity and NPV. Currently, there are more robust studies to support the clinical validity and utility of Afirma GSC and Thyroseq GC when compared to ThyGeNEXT + ThyraMIR.

There is considerable heterogeneity in the methodology and results of various studies of molecular tests. Caution must be applied when interpreting the results of past studies as they often involve older versions of the platforms, and there has been a rapid rate of enhancements of the testing platforms in recent years. It appears that molecular test performance is not meaningfully altered by sonographic risk and that test performance is relatively similar in nodules of various sonographic risk categories. Long-term follow-up studies of outcomes in patients undergoing molecular testing are needed, especially for nonoperated cases with negative test results.

## Figures and Tables

**Table 1 jcm-13-01759-t001:** Comparative overview of commercially available molecular cytology tests in the diagnosis of thyroid nodules.

Characteristic	ThyroSeq GC	Afirma GSC	ThyGeNEXT/ThyraMIR ***
**Methodology**	RNA sequencing DNA sequencing	RNA sequencing	RNA sequencing DNA sequencing microRNA classification
**Number of Genes tested**	NGS DNA and RNA 112 genes (12,135 variants) 120+fusions. Gene expression alterations (19 genes). Copy number alterations (10 chromosomal regions)	GSC: RNA expression analysis of over 10,000 genes XA: 593 genes, 905 variants, and 235 fusion pairs	NGS DNA and RNA 10 genes, 38 fusions, 10 miRNAs
**NPV**	~97%	>90%	~97
**PPV**	~66%	~60%	~75
**Test result categories**	-Negative -Positive (subdivided in subcategories)	-Negative -Suspicious	-Negative -Moderate -Positive
**Can detect specific targetable mutations: BRAFV600E, TERT, RET/PTC, ALK**	BRAF V600E TERT RET/PTC ALK	BRAF V600E RET	BRAF V600E TERT RET/PTC ALK
**Collection process**	1 dedicated pass or diagnostic cytology slides or cell blocks	1 to 2 dedicated passes	1 dedicated pass or diagnostic cytology slides or cell blocks

*** Fewer Number of Studies Available.

## Data Availability

Not applicable.

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
