# Peer review of "Current State of Molecular Cytology in Thyroid Nodules: Platforms and Their Diagnostic and Theranostic Utility"

_jcm, 2024, doi:10.3390/jcm13061759_

Round 1

Reviewer 1 Report

Comments and Suggestions for Authors

1)      In this study, the authors aim to review and compare the performance of the 3 main platforms currently commercially available in the U.S. Comparative studies revealed similar diagnostic performance between these tests. The authors’ idea was useful, and the issue was interesting while the details of the work need some improvements.

2)      As for Afirma GSC (an RNA-based genomic sequencing classifier), could you provide more information of the follicular content index (line 127)?

3)      Line 265-271: exactly as the author mentioned, molecular markers may assist in therapeutic decisions beyond radioactive iodine treatment. Please discuss more of ThyroSeq GC, Afirma GSC and ThyGeNEXT/ThyraMIR in the field of molecular testing, and their contribution, application for the decision-making in clinical practices.

4)      As a new category from the 2022 WHO classification, DHGTC (differentiated high-grade thyroid cancer) was presented with its exclusive characteristics including molecular pattern and transcriptional profiles. Can you please provide the latest data concerning the diagnostic ability of these commercial platforms for DHGTC?

5)      As the author mentioned, ThyroSeq GC, Afirma GSC and ThyGeNEXT/ThyraMIR all possessed high NPV, in avoiding unnecessary thyroidectomy. However, there still lack sufficient data of: a). how the available molecular cytology tests help for further drug use (especially for target therapy); b) follow-up evidence. Please discuss more.

Author Response

  • In this study, the authors aim to review and compare the performance of the 3 main platforms currently commercially available in the U.S. Comparative studies revealed similar diagnostic performance between these tests. The authors’ idea was useful, and the issue was interesting while the details of the work need some improvements.

Thank you very much for the insightful comments and questions. Below are the authors answers.

  • As for Afirma GSC (an RNA-based genomic sequencing classifier), could you provide more information of the follicular content index (line 127)?

The following was added under the line they are referencing here:

The follicular content index measures the relative abundance of follicular cells in the biopsy sample, which can affect the accuracy and interpretation of the GSC result. The specific details of how the FCI is calculated may be proprietary to the Afirma GSC test and may not be publicly disclosed by Veracyte. In general, the FCI is determined using a combination of gene expression patterns associated with follicular cells and other cell types present in the thyroid nodule. The expression levels are then normalized by the expression levels of other reference genes that are not affected by follicular content. The normalized expression levels are then summed to obtain the follicular content index. The higher the follicular content index, the more follicular cells are present in the sample.

3)      Line 265-271: exactly as the author mentioned, molecular markers may assist in therapeutic decisions beyond radioactive iodine treatment. Please discuss more of ThyroSeq GC, Afirma GSC and ThyGeNEXT/ThyraMIR in the field of molecular testing, and their contribution, application for the decision-making in clinical practices.

Thanks for the comments. The following was added under the Therapeutic implications for the future section:

Therapeutic implications for the future

Molecular testing in thyroid nodule FNA has been primarily used for diagnostic purposes. Because of its high negative predictive value, many surgeries can be avoided in patients with indeterminate cytology and negative molecular testing. Recent advances in our understanding of the association between genetic alterations, cancer phenotype, and risk of aggressive behavior have allowed us to use the information provided by molecular tests to further guide management in different aspects:

  • Surgical Planning: For patients who do undergo surgery, molecular testing can provide valuable information for surgical planning. It can help identify the extent of surgery needed, such as whether total thyroidectomy or lobectomy is appropriate, based on the risk of malignancy and the presence of specific genetic alterations.
  • Prognosis: Molecular testing can also provide prognostic information that helps predict the likelihood of recurrence or aggressive behavior of thyroid cancer. For instance, more aggressive tumors, such as differentiated high-grade thyroid cancer (DHGTC), a recently defined category from the 2022 WHO classification, are known to have exclusive characteristics, molecular patterns and transcriptional profiles with higher BRAFV600Emutation and gene fusions   This information is valuable for guiding postoperative management decisions, including the need for adjuvant therapy and the frequency of surveillance.
  • Monitoring Response to Therapy: Molecular testing can be used to monitor response to therapy in patients with advanced thyroid cancer. Changes in molecular profiles over time may indicate treatment response or the emergence of resistance mechanisms, guiding adjustments to treatment strategies.
  • Hereditary Risk Assessment: Molecular tests are able to identify germline mutations associated with hereditary thyroid cancer syndromes. Identification of these mutations allows for appropriate genetic counseling and screening of at-risk family members.
  • Treatment Selection: In cases of advanced or recurrent thyroid cancer, molecular testing may inform treatment decisions, such as the selection of targeted therapies or participation in clinical trials based on the presence of specific molecular alterations. Molecular testing can be used to assess the mRNA expression of sodium iodine symporter (NIS) in the sampled tissue and potentially predict the effectiveness of radioactive iodine (RAI) treatment. Furthermore, molecular markers may assist in therapeutic decisions beyond RAI, although this practice has yet to be widely adopted. 6 The Food and Drug Administration has approved therapeutic options for the treatment of advanced BRAFV600E-mutated thyroid carcinomas, NTRK fusions, RET-mutated medullary thyroid carcinoma, and RET-fusion papillary thyroid cancer. 27  As advances in the field continue to evolve, it is predicted that further treatments targeting specific genetic alterations will continue to become available.

4)      As a new category from the 2022 WHO classification, DHGTC (differentiated high-grade thyroid cancer) was presented with its exclusive characteristics including molecular pattern and transcriptional profiles. Can you please provide the latest data concerning the diagnostic ability of these commercial platforms for DHGTC?

Thanks for the comment.

As mentioned in response to question #3, the following was added under the Therapeutic implications for the future section:

  • Prognosis: Molecular testing can also provide prognostic information that helps predict the likelihood of recurrence or aggressive behavior of thyroid cancer. For instance, more aggressive tumors, such as differentiated high-grade thyroid cancer (DHGTC), a recently defined category from the 2022 WHO classification, are known to have exclusive characteristics, molecular patterns, and transcriptional profiles with higher BRAFV600Emutation and gene fusions   This information is valuable for guiding postoperative management decisions, including the need for adjuvant therapy and the frequency of surveillance.

5)      As the author mentioned, ThyroSeq GC, Afirma GSC and ThyGeNEXT/ThyraMIR all possessed high NPV, in avoiding unnecessary thyroidectomy. However, there still lack sufficient data of: a). how the available molecular cytology tests help for further drug use (especially for target therapy); b) follow-up evidence. Please discuss more.

Please see the suggested addition under comment #3

Wong, K. S., Dong, F., Telatar, M., Lorch, J. H., Alexander, E. K., Marqusee, E., ... & Barletta, J. A. (2021). Papillary thyroid carcinoma with high-grade features versus poorly differentiated thyroid carcinoma: an analysis of clinicopathologic and molecular features and outcome. Thyroid, 31(6), 933-940.

Wong, K. S., Dong, F., Telatar, M., Lorch, J. H., Alexander, E. K., Marqusee, E., ... & Barletta, J. A. (2021). Papillary thyroid carcinoma with high-grade features versus poorly differentiated thyroid carcinoma: an analysis of clinicopathologic and molecular features and outcome. Thyroid, 31(6), 933-940.

Reviewer 2 Report

Comments and Suggestions for Authors

Dear authors,

The manuscript “Current State of Molecular Cytology in Thyroid Nodules: Platforms and their Diagnostic and Theranostic Utility”, jcm-2868511, describes and compares three main commercially available molecular cytology tests for the diagnosis of thyroid nodules: Afirma GSC, Thyroseq GC, and ThyGeNEXT/ThyraMIR. The paper gives a comparative overview of three major platforms useful for thyroid nodule classification, thyroid carcinoma prognosis, and high-risk patient selection. It is written in an easy-to follow way, and with a few minor corrections, I recommend it for publication.

Minor:

1. Despite being mentioned, there isn't much discussion of factors such as cost and processing time among the tests. Please make the comparison and provide a few sentences about it in the text and Table 1.

2. An overview of the number of genes tested (variation evaluated) in each of the tests should be included in Table 1.

3. The frequency of the BRAFV600E mutation varies a lot among studies. It has been reported to be present in about 30-85% of cases of PTC. Therefore, please correct the following sentence in line 80:” Similarly, we know that the BRAF V600E mutation is harbored in about 66% of papillary thyroid cancers”.

4. Line 81 contains the term "Formatting…." I suppose it is a typographical mistake. Please delete it.

5. Lines 124-126: Since parathyroid and medullary thyroid cancer are not mutations, please rewrite the sentence: “When a sample is sent for Afirma testing, it is first run through initial classifiers to identify rare neoplasms and lesions with >95% risk of malignancy (strong driver mutations), such as parathyroid, medullary thyroid cancer, BRAF, and RET:PTC fusion”.

6. I advise avoiding splitting the table.

Author Response

  1. Despite being mentioned, there isn't much discussion of factors such as cost and processing time among the tests. Please make the comparison and provide a few sentences about it in the text and Table 1.

Thanks for the comment, the following has been added under the Comparison of Molecular tests section:

ThyroSeq GC and Afirma GSC have similar processing times of 7 to 10 business days after the sample is received by the laboratory. The processing time for ThyGeNEXT/ThyraMIR may vary, but it usually takes around 2 to 3 weeks to receive results after the laboratory receives the FNA sample. ThyroSeq GC and Afirma GSC costs are comparable at several hundred to a few thousand dollars. The combined cost of ThyGeNEXT (genetic testing) and ThyraMIR (microRNA testing) is typically higher compared to the other tests. It can range from a few thousand to several thousand dollars. These tests are currently widely in use in the United States and almost always covered by commercial insurance. The cost of the tests will depend on the benefits available through insurance.  Deductible, copay, or coinsurance may apply depending on the case. Medicare covers testing. Patients may be eligible for financial assistance programs and can apply through the manufacturer’s website. The tests have been studied for cost-effectiveness since by avoiding unnecessary surgeries in up to 50% of indeterminate cytology nodules; there is far less cost to the health care system from using the tests. Not to mention the potential costs of life-long levothyroxine replacement.

  1. An overview of the number of genes tested (variation evaluated) in each of the tests should be included in Table 1.

Thanks for the comment, The requested information has been added to the table. Also, the following was added in lines 144-145: Afirma Expression Atlas (XA) comprised of 593 genes, 905 variants, and 235 fusion pairs

  1. The frequency of the BRAFV600E mutation varies a lot among studies. It has been reported to be present in about 30-85% of cases of PTC. Therefore, please correct the following sentence in line 80:” Similarly, we know that the BRAF V600E mutation is harbored in about 66% of papillary thyroid cancers”.

Thanks for the comment, the line has been modified as suggested to: Similarly, we know that the BRAF V600E mutation is highly prevalent in papillary thyroid cancers with its reported frequency varying from 30 to up to 85% among studies, 

  1. Line 81 contains the term "Formatting…." I suppose it is a typographical mistake. Please delete it.

We defer this formatting inquiry to the editors.

  1. Lines 124-126: Since parathyroid and medullary thyroid cancer are not mutations, please rewrite the sentence: “When a sample is sent for Afirma testing, it is first run through initial classifiers to identify rare neoplasms and lesions with >95% risk of malignancy (strong driver mutations), such as parathyroid, medullary thyroid cancer, BRAF, and RET:PTC fusion”.

The sentences was modified as follows:

When a sample is sent for Afirma testing, it is first run through initial classifiers to identify if the sample contains parathyroid origin cells or rare neoplasms and lesions with >95% risk of malignancy (strong driver mutations), such as RET mutations causing medullary thyroid cancer, BRAF, and RET:PTC fusion .

  1. I advise avoiding splitting the table.

We defer this formatting inquiry to the editors.

Dharampal N, Smith K, Harvey A, Paschke R, Rudmik L, Chandarana S. Cost-effectiveness analysis of molecular testing for cytologically indeterminate thyroid nodules. J Otolaryngol Head Neck Surg. 2022 Dec 21;51(1):46. doi: 10.1186/s40463-022-00604-7. PMID: 36544210; PMCID: PMC9773581.